# Historical Text Image Enhancement Using Image Scaling and Generative Adversarial Networks

**DOI:** 10.3390/s23084003

**Published:** 2023-04-14

**Authors:** Sajid Ullah Khan, Imdad Ullah, Faheem Khan, Youngmoon Lee, Shahid Ullah

**Affiliations:** 1Multimedia Information Processing Lab, Department of Information and Communication Engineering, Chosun University, Gwangju 61452, Republic of Korea; 2Department of Information System, Prince Sattam bin Abdulaziz University, Al-Kharj 16278, Saudi Arabia; 3Department of Computer Engineering, Gachon University, Seongnam-si 13120, Republic of Korea; 4Department of Robotics, Hanyang University, Ansan-si 15558, Republic of Korea; 5Faculty of Engineering, University Malaysia Sarawak, Kota Samarahan 94300, Malaysia

**Keywords:** text image enhancement, wavelet transform, generative adversarial network, machine learning

## Abstract

Historical documents such as newspapers, invoices, contract papers are often difficult to read due to degraded text quality. These documents may be damaged or degraded due to a variety of factors such as aging, distortion, stamps, watermarks, ink stains, and so on. Text image enhancement is essential for several document recognition and analysis tasks. In this era of technology, it is important to enhance these degraded text documents for proper use. To address these issues, a new bi-cubic interpolation of Lifting Wavelet Transform (LWT) and Stationary Wavelet Transform (SWT) is proposed to enhance image resolution. Then a generative adversarial network (GAN) is used to extract the spectral and spatial features in historical text images. The proposed method consists of two parts. In the first part, the transformation method is used to de-noise and de-blur the images, and to increase the resolution effects, whereas in the second part, the GAN architecture is used to fuse the original and the resulting image obtained from part one in order to improve the spectral and spatial features of a historical text image. Experiment results show that the proposed model outperforms the current deep learning methods.

## 1. Introduction

Document Analysis and Recognition (DAR) research is highly focused on protecting and improving the readability of historical text images. Historical documents are an invaluable part of our cultural heritage, having a significant scientific and cultural value. Historical documents typically contain important information and are extremely valuable [1]. The digitization of ancient texts is an essential step toward protecting the literature. However, manual processing of these large volumes of documents takes time and effort. As a result, it is necessary to use cutting-edge image processing and machine learning methods to process historical text documents automatically. Recent text documents can easily be enhanced. However, due to degradation such as noise, fading, artifacts, molds, water stains, stamps, contrast, brightness, and so on, historical text images are much more difficult to enhance. These distortions may have severe impact on post-processing steps such as Optical Character Recognition (OCR), segmentation, indexing, and information spotting.

The Convolutional Neural Network (CNN) model has recently gained popularity in DIE-related subtasks such as binarization [2,3], de-fading [4], and de-blurring [5]. Despite the fact that CNN and its family of algorithms outperform previous image processing techniques, they have some flaws. CNNs restore different regions using the same convolution filters, resulting in spectral information loss. Additionally, CNNs fail to extract low-level spatial features from text images. 

With the advent of deep learning (DL), several research studies applied cutting-edge deep-learning-based models to various image processing and computer vision tasks, such as image enhancement [3,6,7], object detection [8], image text segmentation [9], etc. DL methods have been shown to deliver promising results and outperform traditional methods. Likewise, in recent years, DL-based methods for text image enhancement have received a lot of attention. 

The motivation behind this study is to present a hybrid image processing and DL-based method to enhance historical text images. Our contributions are twofold. First, the transformation method is used to reduce noise and blurring effects while increasing intensity resolution. A new bi-cubic interpolation of Lifting Wavelet Transform (LWT) and Stationary Wavelet Transform (SWT) is employed to enhance image resolution and to reduce noise and blurring effects. Second, the GAN architecture is employed to extract the spatial and spectral features of a historical text image. 

The rest of the paper is organized as follow: Section 2 introduces the related work on text image enhancement. Our proposed method is explained in Section 3. Section 4 illustrates qualitative and quantitative experiments and provides the discussion, and Section 5 presents the conclusion and future directions. 

## 2. Related Work

Text in historical documents is difficult to read due to degradation. The Document Analysis and Recognition (DAR) community has paid much attention to text image enhancement. Documents enhancement problems consist of several tasks such as binarization, de-blurring, de-noising, de-fading, watermark removal, and shadow removal. This research work is aimed to solve the problem of image de-blurring, image de-noising, and image de-fading. Figure 1 visually represents these problems.

Similarly, different datasets are available for text document enhancement tasks. Table 1 illustrates the specification of these datasets.

There are numerous challenges associated with historical text image enhancement, such as text enhancement without affecting the background because the background contains important information or objects that must be preserved. Some higher-resolution images because the algorithm to slow down and we need a memory-efficient algorithm to process the image. A flash light is now an essential part of image acquisition. When a flash strikes on an object, the light reflection distorts the image and reduces readability. Furthermore, various algorithms present ideas on how to improve the image’s edges and curves, allowing for enhanced text. Text images with low resolution require enhancement for readability as well, but we must first improve the image’s resolution.

A simple thresholding method was first used for text image enhancement. In this method, a threshold value was used to segment or to classify the background and foreground images [10,11]. These techniques have been progressing in the past few years, by using machine learning algorithms such as support vector machines (SVM) [12]. Subsequently, methods based on energy were presented to monitor the text pixels by optimizing their energy function and minimizing the effect of the degraded image background. However, the results obtained using those methods were unsatisfactory [13,14].

An idea was introduced in [15] proposing a method to enhance the contrast of the image by preventing the over-enhancement problem. In this method, the original image was decomposed using Discrete Wavelet Transform (DWT), and then the CLAHE technique was applied to the low-frequency component. Finally, weighted average was taken from the original and the resultant image to obtain an enhanced image. However, this method is not suitable for grayscale images and fails in the image de-fading task. 

Using deep learning approaches, the problems of image de-blurring, de-fading, and de-nosing were resolved by learning the enhancement directly from the original data. A pixel’s classification method was proposed in [16] to classify each pixel as white or black depending on the neighboring pixels. For this task, a 2D long- and short-term memory was trained. However, it is a time-consuming process.

Convolutional Neural Network (CNN) auto-encoders provide a more practical solution for image de-noising. As a result, they have become the leading edge of recent advances in image de-noising [17] and, more specifically, in text image enhancement operations such as de-blurring [5] and image binarization [2]. 

A full CNN model was suggested in [18] to binarize deteriorated images at various image scales. Likewise, a pre-trained U-Net model was presented in [19] to learn binarization with less data. A conditional generative adversarial network (GAN) was introduced in [20] for various image enhancement tasks such as binarization, de-blurring, and watermarks removal. A modified version of [20] was presented in [21] by using dual discriminator to evaluate text readability. Similar cGAN’s techniques were presented in [22,23,24]. 

Recently, transformer-based architectures were used in few papers for natural image restoration and image de-warping [25]. However, the architectures that were used kept relying on the CNN method before moving to the transformer stage. A Document Image Transformer (DocTr) was proposed in [26] to solve the problem of geometry and illumination distortion of the document images. In this method, the global context of the document image was captured and pixel wise detection method was used to correct the distortion. Another method was proposed in [27] quantify the downstream analysis effects using empirical analysis. In this study, bais-variance decomposition of an empirical risk function and mean square error were measured using bootstrap data simulation. Again in [28] provides a report on the methodology and performance of the five submitted algorithms by various research groups across the world. Binarization and Classification were done using well know evaluation metrics. Similarly, in [29], a software framework, Binarization Shop, that combines a series of binarization approaches that have been tailored to exploit user assistance was presented. This method provides a practical approach for converting difficult documents to black and white. However, it was a time consuming process and takes long time for training. Contrarily, we proposed a novel model to reduce noise and blurring effects with increased pixels resolution. A new bi-cubic interpolation of Lifting Wavelet Transform (LWT) and Stationary Wavelet Transform (SWT) is employed to enhance pixels resolution and to reduce noise and blurring effects. Second, the GAN architecture is employed to extract the spatial and spectral features of a historical text image.

## 3. Materials and Methods

### 3.1. Data Collection

Four datasets were used in this study. These datasets are Noisy Office [30] database, S-MS (Synchromedia Multispectral Ancient Document Images) [27], Bishop Bickley diary [29], Tobacco 800 [31], and Blurry document Images. These datasets are also publicly available online. The details are given in Table 1.

**Table 1 sensors-23-04003-t001:** Datasets used for image enhancement tasks.

Database	Task	Number of Images	Real/Synthetic	Image Size (Resolution)
Noisy Office [30]	Image de-noising	288	Mixed	Variable
S-MS [27]	Mixed (de-noising, de-blurring, de-fading)	3.4309	Synthetic	1001 × 330
Bishop Bickley diary [29]	Image binarization	07	Real	1050 × 1350
Tobacco 800 [31]	Image de-noising	1290	Real	1200 × 1600
Smart Doc-QA [32]	Image de-blurring	4260		351 × 292
Blurry document images [5]	Image de-blurring	35,000	Synthetic	300 × 300

### 3.2. Methodology

The proposed method is divided into two steps, which are explained below.

#### 3.2.1. Pixels Resolution Enhancement

##### LWT Bi-Cubic Interpolation Method

In this part, the primary concept which is being followed to enhance resolution is wavelet transformation of the image. Three key concepts, Lifting Wavelet Transform (LWT), Stationary Wavelet Transform (SWT), and image interpolation, altogether sum up to contribute to the resolution enhancement. Initially, the input image is decomposed using a one-level LWT. The Daubechies transformation approach is used to decompose the image which is empirically determined for obtaining the best results. After decomposition, vertical, horizontal, diagonal, and approximation coefficients are obtained denoted by LH, HL, HH, and LL. Figure 2 shows the LWT coefficients representation. 

Initially, the input image with resolution m×n is subjected to the LWT function which yields three high-frequency sub-bands, i.e., horizontal, vertical, and diagonal details of the image as shown in Figure 2. We chose the level of decomposition as “1”, because it produces good results. The only one low-frequency sub-band contains approximation coefficient. Then the high-frequency sub-bands are processed using bi-cubic interpolation process. The interpolation function is chosen as “2”. It zooms the image without decreasing the resolution. Figure 3 depicts the effects of bi-cubic interpolation.

LWT decomposes the m × n band into m × n/2 sub-bands. The interpolation yields the resultant sub-bands with m × n resolution. 

##### Stationary Wavelet Transform (SWT)

LWT down-samples the sub-bands so as to generate high-frequency components, which cause information loss. To mitigate this loss, SWT is employed to minimize the down-sampling loss. The input image undergoes SWT and yields four sub-bands as generated by LWT. The sub-bands generated by SWT are of m × n resolution comprising of horizontal, vertical, and diagonal details. 

##### Addition of Sub-Bands and Interpolation

The high-frequency sub-bands obtained from LWT and SWT are summed together. These sub-bands include interpolated H, V, D from LWT and H, V, D from SWT, such that the corresponding components are summed together. The resultant sub-bands obtained after adding the corresponding sub-bands from LWT and SWT are then subjected to further interpolation by some factor “*γ*”. Simultaneously the original image also undergoes interpolation by factor “*γ*”.

##### Inverse Lifting Wavelet Transform

After obtaining sub-bands by adding corresponding LWT and SWT sub-bands and applying an interpolation function, the inverse LWT function is applied to the interpolated original image and resulting sub-bands. A new image is obtained with the resolution of 2*γ*(m × n). Figure 4 shows the resolution enhancement architecture. 

When the image is up-sampled, it surely loses its quality because of missing pixels. With bi-cubic interpolation, the image is filled with neighboring pixels and resized. In this method, the strength of bi-cubic interpolation and strengths of wavelet function ensure that the image is scaled up and the quality of the image is preserved. This method enhances the quality without the staircase effect and reduces noise, blurriness, and jaggedness.

#### 3.2.2. Generative Adversarial Network (GAN)

The purpose of the GAN model is to extract the spatial and spectral features of a historical text image. It is employed to fuse the original and the resulting image achieved from part one to improve the spectral and spatial features of a historical text image. GAN is made up of two modules: generator and discriminator. The prime objective of GAN is to generate data patterns that are similar to the input data. The goal of the discriminator is to distinguish the resulted image from the source image. When the generator fools the discriminator, it indicates that the generator is well trained. Following the application of the GAN model, the resulting fused image contains all of the complementary information present in the input images, namely the fused image obtained from part 1 and the original image. Figure 5 depicts the entire image fusion process. 

Initially, the input original and fused (obtained from part 1) images are given to generator. Then, fused and input images are compared. This approach generates a minimum/maximum game between the GAN modules. The generated fused image is again sent to discriminators. The discriminator receives the fused image and the original image as an input. 

The generator uses a five-layer CNN with 3 × 3 filters in each layer. Each of these five convolution layers has a stride value of “1”. Convolution layers are used to extract feature maps from an input image. In generator architecture, batch normalization is used to make the model more stable. In all layers of the generator, “*Leaky Relu*” is employed except the last layer for activation. In the last layer, “*tanh*” function is used. The “*tanh*” and “*Leaky Relu”* functions are expressed in Equations (1) and (2).
(1)LkyRlu=max(0.1x,x),
(2)tnh=tnh(x)

The discriminator is made up of five layers of CNN with 3 × 3 filters. The final filter has only one 1 × 1 filter. Each of these five convolution layers has a stride value of “2”. In discriminator architecture, batch normalization is used between the first and the last layers. In all layers of the generator, “*Leaky Relu*” is employed except the last layer. Similar to the generator, in all layers of the discriminator, “*Leaky Relu*” is employed except the last layer for activation. In the last layer, “*tanh*” function is used.

## 4. Results

The experiments have been carried out, and it has been found out that our proposed method outclasses the existing state-of-the-art methods. All experiments were conducted using MATLAB 2021 installed on a Core i7 (9th Generation) processor with 1.9 GHz and 16 GB of RAM. To validate our proposed method, the datasets mentioned in the Materials and Methods section are used. The quantitative and qualitative results show significant performance as compared to the existing methods. We have noticed the superiority performance of our proposed method in recovering highly degraded historical text images. We used the same criteria in our experimental results as in the existing methods.

The performance assessment metrics employed in this study are F-measures (FM), Peak Signal-to-Noise Ratio (PSNR), Word Error Rate (WER), and Distance-Reciprocal Distortion Metric (DRD). Figure 6, Figure 7 and Figure 8 show the qualitative results of our proposed and the existing deep learning methods published in 2021 and 2022. 

Figure 6, Figure 7 and Figure 8 clearly depict the superiority of our proposed algorithm. However, the experimental results may not be perfect all the time, especially where the input image has a light text and dull background. During the experiments, it was observed that the resulting image from [2] was de-blurred well, but it was contaminated with background noise. However, its performance was satisfactory. The output achieved form [3] was free from noise but had blurring effects. Our proposed method initially de-blurred and de-noised the image with enhanced pixels resolution. Using the bi-cubic interpolation function, the input image was zoomed without any distortion. 

Almost in all cases, the results achieved using S. Kang’s approach [2] are satisfactory and corporately higher than those of all other methods. This approach even still provides some better results as compared to our proposed method in terms of FM. This is because some images available from Noisy Office and Images datasets have dull backgrounds. S. K. Jemni’s [3] results are satisfactory; however, the high brightness causes some text distortion. Similarly, the proposed bi-cubic interpolation of LWT and SWT has better performance but with lower contrast. The application of GAN architecture enhances the visual quality due to complementary information transformation from input to output of images without distortion. GAN has the ability to generate equal distribution of frequencies received from input images. If we look at the final images in Figure 6, Figure 7 and Figure 8, the intensity variations are relatively optimal. The VID is much higher than for the rest of the methods due to the application of the GAN architecture, as it transforms the complementary information with any distortion of the text to the final fuse image. 

Additionally, we have performed quantitative analysis to further evaluate our proposed method. Table 2, Table 3 and Table 4 illustrate the quantitative results of our proposed and the comparative methods. In few cases, when the images are very dull, our algorithm fails to enhance the input image and the images lose their details.

Table 2, Table 3 and Table 4 depict that our proposed algorithm achieved maximum results except in few cases. In the case of Noisy Office dataset, S. Kang [2] achieved higher FM than that of our proposed method due to few images with dull backgrounds. For the mixed dataset, the WER result of S. K. Jemni [3] is better than that of S. Kang [2] and shows competitive performance.

PSNR is most widely used as a quantitative metric which provides a pixel-wise evaluation and shows the effectiveness of document enhancement method in terms of visual quality. The primary objective of the first part of our proposed method is to enhance intensity variations, which increases the PSNR rate. Similarly, with the usage of the GAN architecture, the WER and DRD rates increase dramatically due to complementary information transformation from input to output of images without distortion. 

To further evaluate our proposed method, images from datasets were analyzed and the results showed that our proposed method achieved satisfactory results. Table 5 and Table 6 illustrate the quantitative results of our proposed method and the existing comparative methods. 

It was observed from Table 5 and Table 6 that our proposed algorithm achieved maximum results except in the case of WER for “Images” dataset due to some images having dull backgrounds, and because our method could not de-blur them well. For Tobacco dataset, our proposed method shows better performance. 

## 5. Conclusions and Future Direction

This article proposed a new method towards the improvement of historical text images. Initially, the image is scaled up using a new bi-cubic interpolation of Lifting Wavelet Transform (LWT) and Stationary Wavelet Transform (SWT) to enhance pixels resolution and to reduce noise and blurring effects without degrading the quality of the image. Second, the GAN model is employed to extract the spatial and spectral features of a historical text image. The proposed method may be implemented in future for X-ray and CT images to assist clinical process. There is, however, still room for improvement.

Text readability is improved by sharpening the edges and lowering the contrast of the image’s background. 

Overexposure and underexposure correction, super resolution, and OCR performance evaluation are a few key areas open for future work. Overexposure arises when excessive light is captured during the digitization process, particularly when using a smart phone. The camera flash adds an excessive amount of reflection or glare to the image, and as far as we know, no study is available to overcome this issue for historical text image enhancement. The same is true for underexposure when the lighting is poor. The underexposure problem received high attention recently. Moreover, fading is also a problem of attention and under-studied task. It occurs due to aging, washing out and illumination, etc. 

Super-resolution of documents is a burning area in the field of image analysis and understanding. Low-resolution images create uncertainties for character recognition methods. Super-resolution is more challenging, especially when historical documents have noises and artifacts. In the future, effective deep learning methods, particularly transformer-based super-resolution methods, will be required to improve OCR performance.

Enhancing Optical Character Recognition, or OCR, is another crucial subject that needs attention in order to help automated text analysis of documents. Existing practices either overlook assessing their techniques in terms of OCR, or reveal OCR improved performance only on a small number of images, which is insufficient to demonstrate the practicality of their methodologies. Appendix A show the examples of these tasks in the Appendix A. Appendix A depicts an interpolation example.

## Figures and Tables

**Figure 1 sensors-23-04003-f001:**
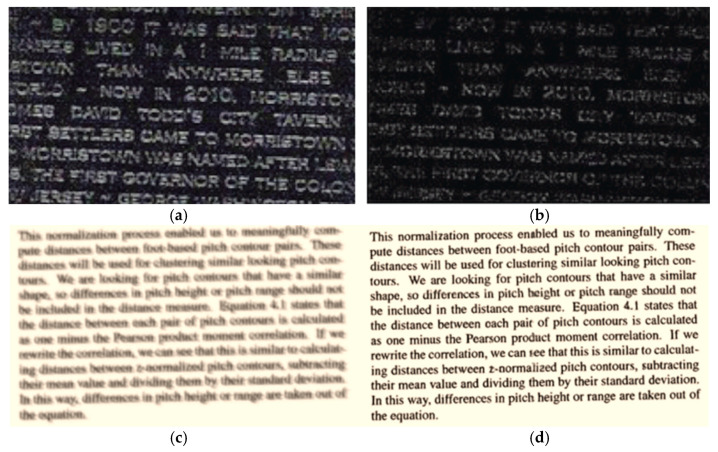
Visual representation of text image enhancement tasks: (**a**,**c**,**e**) input images, (**b**) de-noised image, (**d**) de-blurred image, (**f**) de-faded image.

**Figure 2 sensors-23-04003-f002:**
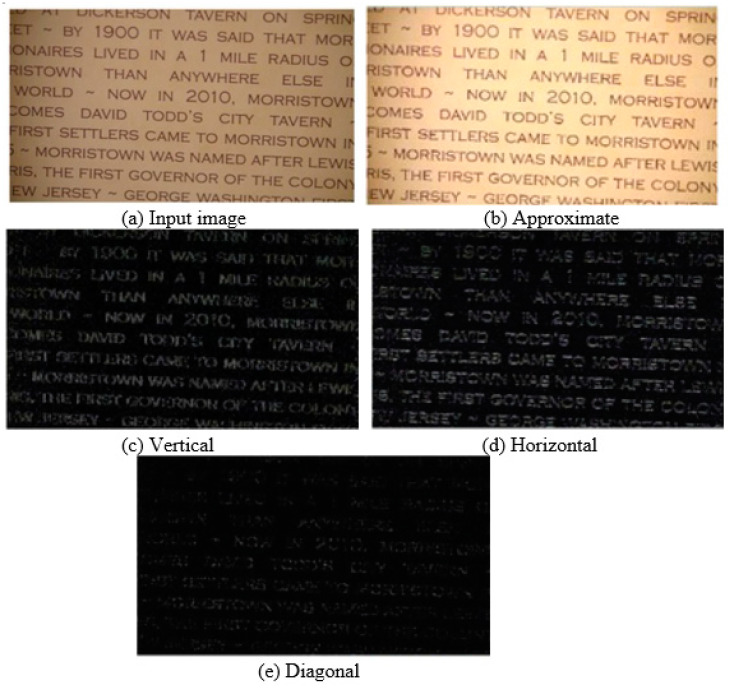
LWT coefficients.

**Figure 3 sensors-23-04003-f003:**
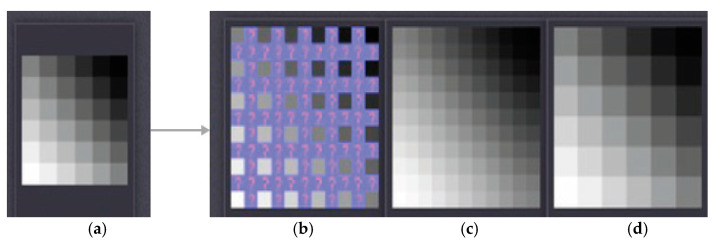
Bi-cubic interpolation effects: (**a**) input image, (**b**) before interpolation, (**c**) after interpolation, and (**d**) no interpolation.

**Figure 4 sensors-23-04003-f004:**
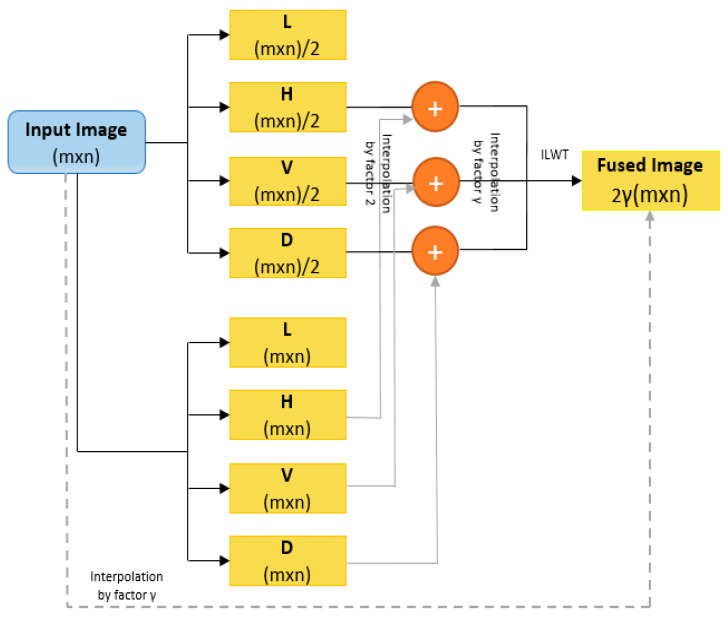
Intensity resolution.

**Figure 5 sensors-23-04003-f005:**
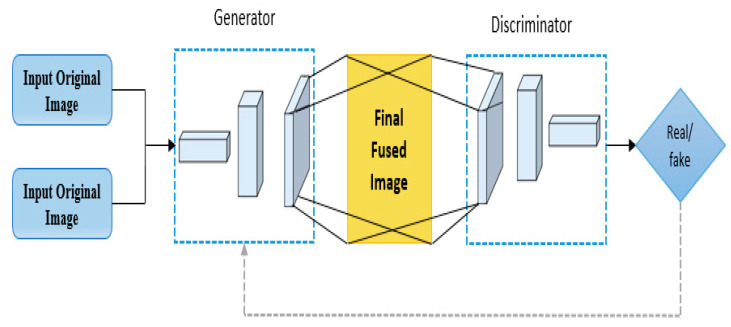
GAN fusion process.

**Figure 6 sensors-23-04003-f006:**
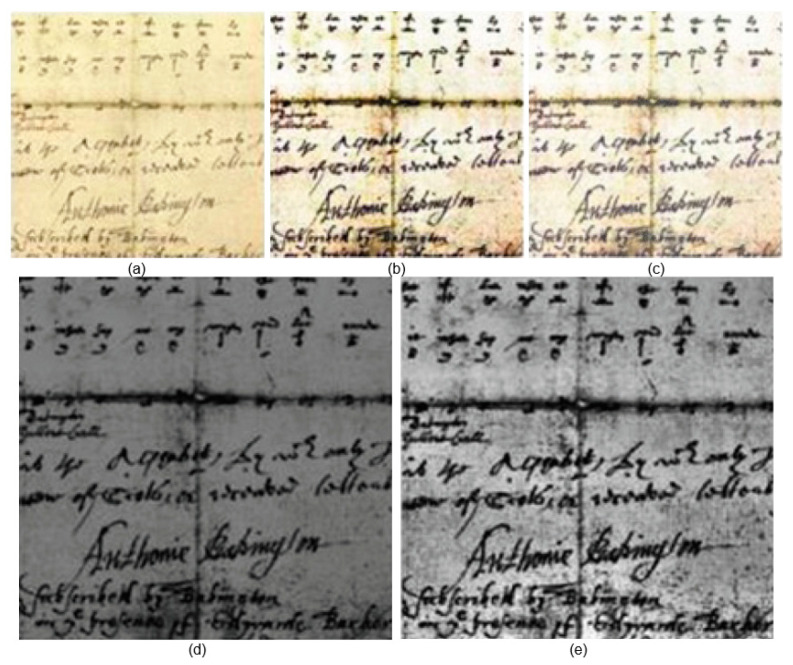
Qualitative results using Noisy Office dataset [30]: (**a**) input image, (**b**) S. Kang [2], (**c**) S. K. Jemni [3], (**d**) proposed LWT and SWT bi-cubic interpolation result, (**e**) proposed final image.

**Figure 7 sensors-23-04003-f007:**
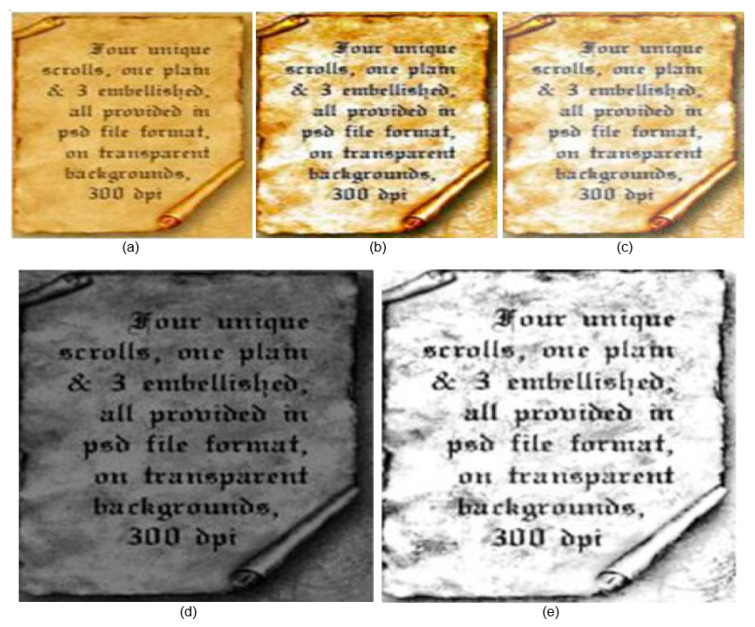
Qualitative results S-MS [27]: (**a**) input image, (**b**) S. Kang [2], (**c**) S. K. Jemni [3], (**d**) proposed LWT and SWT bi-cubic interpolation result, (**e**) proposed final image.

**Figure 8 sensors-23-04003-f008:**
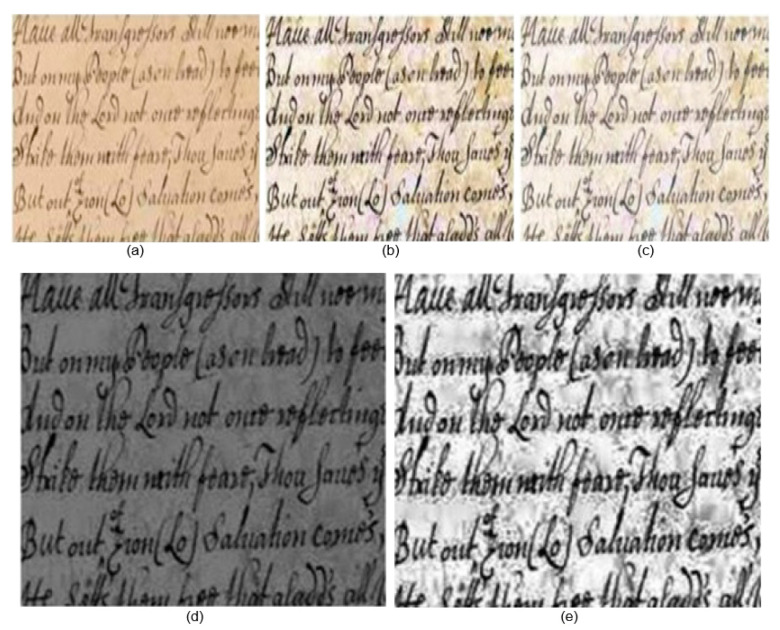
Qualitative results using Blurry document images dataset [5]: (**a**) input image, (**b**) S. Kang [2], (**c**) S. K. Jemni [3], (**d**) proposed LWT and SWT bi-cubic interpolation result, (**e**) proposed final image.

**Table 2 sensors-23-04003-t002:** Comparative results on Noisy Office [30] dataset.

Method	PSNR	FM	WER	DRD
S. Kang [2] (CNN)	14.42	**96.54**	85.45	1.23
S. K. Jemni [3] (_C_GAN)	14.09	94.21	84.95	1.18
LWT &SWT	13.95	94.18	83.12	1.01
Final Proposed	**15.02**	96.51	**86.17**	**1.31**

**Table 3 sensors-23-04003-t003:** Comparative results on S-MS [27] dataset.

Method	PSNR	FM	WER	DRD
S. Kang [2] (CNN)	17.31	85.61	74.21	2.02
S. K. Jemni [3] (_C_GAN)	16.99	85.21	74.19	1.99
LWT and SWT	16.91	84.02	73.91	1.88
Final Proposed	**18.01**	**85.71**	**73.32**	**2.44**

**Table 4 sensors-23-04003-t004:** Comparative results on Blurry document images dataset [5].

Method	PSNR	FM	WER	DRD
S. Kang [2] (CNN)	19.42	98.12	81.28	2.11
S. K. Jemni [3] (_C_GAN)	19.02	97.84	81.31	2.10
LWT and SWT	18.21	94.02	80.21	2.01
Final Proposed	**19.49**	**97.13**	**82.12**	**2.17**

**Table 5 sensors-23-04003-t005:** Comparative results on Tobacco 800 [31] dataset.

Method	PSNR	FM	WER	DRD
S. Kang [2] (CNN)	21.02	93.21	83.21	2.18
S. K. Jemni [3] (_C_GAN)	20.51	93.01	83.02	2.07
LWT and SWT	19.79	92.81	82.88	1.98
Final Proposed	**21.74**	**93.24**	**83.84**	**2.57**

**Table 6 sensors-23-04003-t006:** Comparative results on S-MS [27] dataset.

Method	PSNR	FM	WER	DRD
S. Kang [2] (CNN)	19.42	**87.12**	79.51	1.87
S. K. Jemni [3] (_C_GAN)	18.91	86.74	79.11	1.59
LWT and SWT	18.02	85.07	78.93	1.02
Final Proposed	**19.77**	87.05	**79.59**	**1.91**

## Data Availability

The datasets generated and/or analyzed during the current study are publicly available as mentioned in the Section 3. However, datasets may be provided by the corresponding author on reasonable request.

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
