# Peer review of "Historical Text Image Enhancement Using Image Scaling and Generative Adversarial Networks"

_sensors, 2023, doi:10.3390/s23084003_

Round 1

Reviewer 1 Report

The authors have proposed a hybrid method to enhance the quality of historical text images. This method has two parts: the first part uses the transformation method and LWT, and SWT to enhance 58 pixels resolution, and reduce the noise; The second part adopts the GAN to extract the spatial and spectral features of the historical text image. But, some issues should be further considered. 1. In the abstract, and the introduction section, the aim of the GAN is not clear because the authors only discribe they use the GAN to extract the spatial and spectral features, but what does it do after extraction, what effect does it produce. 2. The content in page 10 should be above the Section 5; 3. The experimental results should be further analyzed, especially the reasons for the slightly lower performance in Tables 2 and 4. 4. Why the authors not directly use the GAN to enhance the quality of the text images. The reasons behind the first part should deeply analysed.

Author Response

See Reviewer 1 attached file

Reviewer 2 Report

Good Work done. yet, Grammatical errors to be checked in many sentences. (e.g) line 213, "Experiments has been....."

Fig. 4 caption is missing.(line number 179)

In Line 194, Part 1 mentioned is not shown in Fig. 5.

In many places, it is written as "above", "below". To be avoided and replaced with section number, figure number etc. (line no. 138, 234,278)

line number 251 and 252 continuity missing

Author Response

See Reviewer 2 attached file

Reviewer 3 Report

To enhance for text image recognition and analysis tasks, a new bi-cubic interpolation of lifting and stationary wavelet transform is proposed to enhance pixel resolution. Then application of Generative Adversarial Network (GAN) is used to extract the spectral and spatial features in historical text images.

1. Figure 1 does not correspond to the signal and it is suggested that the position can be adjusted in the relevant software, and Figure 4 is not explained in title.

2. The descriptions of picture 6.7.8 in the text are not sufficient, and it is suggested to add picture descriptions and comparisons.

3. The performance evaluation indicators are FM, PSNR, WER and DRD, what is the reason to be better than other indicators.

4. The experimental part of the paper uses the GAN structure to extract the spatial and chronological spectral features of historical texts, which are described too little.

5. The English used is not professional. It is suggested that the English used is improved.

Author Response

See Reviewer 3 attached file

Round 2

Reviewer 1 Report

The authors have been addressed my questions. But, the quality of English used should be improved.

Author Response

Check our paper with proof reader and fixed many grammatical and spelling mistakes. 
